# Risk Communication in the Alert Phase of the COVID-19 Pandemic: Analysis of News Flow at National and Global Levels

**DOI:** 10.3390/ijerph19159588

**Published:** 2022-08-04

**Authors:** Hua Guo, Jiandong Zhang, Shihui Feng, Boyin Chen, Minhong Wang

**Affiliations:** 1Business School, Hohai University, Nanjing 211100, China; 2Faculty of Education, The University of Hong Kong, Hong Kong, China; 3Department of Educational Information Technology, East China Normal University, Shanghai 200062, China

**Keywords:** risk communication, news flow, global media, COVID-19 pandemic, social network analysis

## Abstract

This study examined the global media citation network of COVID-19-related news at two stages of the pandemic alert phase, i.e., the national level alert stage and the global level alert stage. The findings reveal that the small-world pattern and scale-free property of media citation networks contributed to the rapid spread of COVID-19-related news around the world. Within the networks, a small number of media outlets from a few countries formed the backbone of the network to control the risk communication; meanwhile, many media of geographical and cultural similarities formed cross-border collaborative cliques in the periphery of the network. When the alert phase escalated from the national level to the global level, the network demonstrated a number of changes. The findings contribute to the understanding of risk communication for international public health emergencies by taking into account the network perspective and evolutionary nature of public health emergencies in analysis.

## 1. Introduction

In the last two years, the news about the COVID-19 pandemic has played a critical role in the risk communication of the pandemic, not only in the public health domain [1,2] but also in social and economic areas that have been affected by the pandemic [3,4]. Global-wide sharing of the outbreak information through media can be regarded as a collective coping mechanism for public health emergencies [5]. It is important to investigate how risk information is communicated through international news flow, identify the origins of pandemic-related information, and unveiling the law of information dissemination in public health emergencies.

Existing studies on international news flow for risk communication were often conducted based on the data from a limited number of countries [6,7]. They tend to ignore the collaborative nature of global media systems, that is, different countries and their respective news media often interact and generate an interconnected network in crisis and emergency risk communication in a pandemic [8]. They assumed that the news often flows from large and powerful countries to small and peripheral ones [9,10]. Moreover, these studies paid inadequate attention to the evolutionary process of public health emergencies in analyzing related news flow.

The World Health Organization [11] divided the influenza pandemic into four phases: the *interpandemic* phase (the period between influenza pandemics), the *alert* phase (when influenza caused by a new subtype has been identified in humans), the *pandemic* phase (the period of global spread of human influenza caused by a new subtype), and the *transition* phase (when the assessed global risk reduces with de-escalation of global actions).

This study focused on examining the international news flow for risk communication in the *alert* phase of the COVID-19 pandemic. In this phase, the information about the COVID-19 outbreaks is spread throughout the world through collaborative media, allowing people and governments in different countries to perceive the risks of varying severity at different times [1,12,13]. To address the gaps in existing research, this study considered the collaborative nature of global media systems by exploring the global media citation network of COVID-19-related news. Further, considering the evolutionary nature of the pandemic, this study divided the alter phase into two stages to examine how the news flow changed when the alert phase escalated from the national level to the global level.

### 1.1. Risk Communication in Public Health Emergencies

The media always play an important role in the social perception of risk [14,15]. With the ability to quickly disseminate information to a large number of consumers from multiple sources, the news media have been an important channel for the public to seek risk-related information [5,16,17]. In public health emergencies of international concern, media coverage is critical [18], which has the potential to influence people’s perceptions and attitudes about the issues that are involved and ultimately influence their response to a crisis in ways that may have immediate and long-term effects on society [19,20,21,22]. It can even serve as a primary power compared with the spread of the disease to draw the public’s attention and lead their reaction [16].

Discussions about risk are usually time-bound, geographically specific, and relevant to some but not necessarily all populations [5]. Media coverage of public health emergencies is distinguished from regular news coverage in its information extensity, urgency, relevance to the public, and the scope of the issues that are involved [19]. For example, Laing’s [23] study of media during Canada’s pH1N1 influenza A outbreak reported two peaks in coverage, with one related to the arrival of the virus in Canada and the other to the rollout of the immunization campaign. Moreover, there is a lack of journalists that are specialized in crisis coverage across countries. Even if the crisis was international, some media outlets might not assign any journalist [24].

Alvarez-Galvez et al. [25] suggested five determinants of information during disease outbreaks: information sources, structure and consensus of the online community, communication channels, message content, and context. In recent studies, researchers have focused on: (1) the content of news, such as the spread and identification of fake news [26,27] or topics of news coverage [28]; and (2) the perception of risk, such as media coverage that exacerbated public fear and panic about the infectious disease [29,30] or changes in people’s perception of an emerging infectious disease [31]. However, there is a lack of research on the collective actions of global media systems. Yan and Bissell [32] noted that severity is the only consistent determinant of disasters reports and that geographic distance and relevance between countries did not predict any variance of news coverage. Gabore [7] analyzed the news reports of foreign media on the prevention of COVID-19 in Africa and found that Western and Chinese media have significant differences in the use of information sources. Caldarelli et al. [6] studied the flow of online misinformation during the peak of the COVID-19 pandemic in Italy, examining the information-sharing behavior of political subcommunities. These studies inspired our research on the global media collaboration in the COVID-19 pandemic’s alert phase.

### 1.2. International News Flow

World-systems theory views countries as components of a broader global system, and categorizes countries into three interactive zones, namely core, semi-periphery, and periphery [33]. Economically and politically powerful countries (e.g., the U.S. and Western European countries) tend to form the core. In contrast, periphery countries have weak economies and often rely on agriculture. Semi-periphery countries embody the traits of core and periphery states and fall in between when it comes to economic, political, and cultural factors [34]. Based on this theory, news communication reflected the unequal power relations of information in the world. Kim and Barnett [35] were the pioneers to apply the world-systems theory in the analysis of news flow. They pointed out that the strengths of interdependence between the core countries are much stronger than those between the core and the peripheral countries, and the news exchange within the core marginalized the periphery. Chang et al. [36] pointed out that due to structural boundaries between countries, news markets created implicit and invisible parameters that limited the way commercial organizations behave and transact, which kept international news flow in a relatively closed state.

Studies in international news flow commonly used the method of social network analysis (SNA). Part of these studies focused on specific news topics [37], such as media agendas in international disputes [38] and sustainable issues [39]. Other studies tracked the relative prominence of countries through an aggregation of news over a period of time and over a set of media [37]. For example, Fracasso et al. [40] examined the systemic factors in foreign media coverage by using a dataset of the news coverage from 148 European Union national media. Some studies combined the aspects above. One example of such was Blondheim et al.’s [41] study, where they investigated factors for news coverage and predicted trends for news development in the 2009–2012 global economic crisis.

In terms of media attention, Segev [42] suggested that foreign news coverage was greatly impacted by national traits (e.g., the size and power of the foreign country), relatedness (i.e., proximity to that foreign country), and events (e.g., natural disasters, wars, or conflicts). Segev [10] compared world news coverage across countries and source types and found that media coverage of foreign news differed significantly across regions. Specifically, western news sites tend to cover a wide range of foreign news from more countries compared to Asian news sites. In terms of media influence, Himelboim [43] analyzed the use of hyperlinks in news sites and found that the US and the UK were at the center of international news media networks 11 years ago. Guo and Vargo [8] analyzed data from 4708 online news sources, including emerging and traditional media and they confirmed that American media remained the most prominent and had the biggest agenda-setting influence on international news flow.

### 1.3. Global Media Systems

In the complex media circumstance, news production has witnessed changes in its structure, work division, and foci [44]. Moreover, global media systems have become more networked and globalized [45]. News organizations have formed an ecosystem [46], which includes a wide range of news and information providers with news information flowing among them [44]. Due to a limited budget [47] and a lack of sufficient expertise [48], growth in intensity of cross-media partnerships such as cross-production, cloning, cooperation, content sharing, and full convergence is of great significance [47]. Media outlets select information sources based on whether formal cooperation, similar political views, and ease of access exist between themselves and their partners [49,50]. Media ownership is another issue to be concerned about [51]. The complex interaction process of communication strategies and media discourses between the receivers and transmitters has become a key area of risk communication research in the COVID-19 era [14].

Previous research on the structural characteristics of the global media system was built on aspects of complex networks such as small-world and scale-free properties [52]. Gong and Xiao [53] used a small-world network modeling approach to simulate enhanced diffusion effects of epidemic news dissemination. By analyzing the data from June 2014 to March 2015 from six German news sources, Spitz and Gertz [54] found that the inherent backbone of online news article networks was highly sparse and lightweight. However, many studies focused on the communication network structure from the perspective of entity co-occurrence, instead of international news flow. Traag et al. [55] argued that the co-occurrence of people in traditional newspaper media deviated significantly from what would be expected in a random graph. By analyzing the co-occurrence of people’s names in 3000 news articles that were related to economics and politics from the 1987 Reuters newswire, Özgür and Bingol [56] found that the network of the news articles presented the small-world property and low degree of power distribution. More empirical research is needed to investigate the network structure of global media systems.

### 1.4. The Present Study

This study aimed to address the gap in existing research by considering the collaborative nature of global media systems and the evolutionary nature of public health emergencies in analyzing the news flow for risk communication in the COVID-19 pandemic. We explored the global media citation network of COVID-19-related news in the *alert phase* of the pandemic. Moreover, we divided the alter phase into two stages to examine how the news flow changed when the alert phase escalated from the national level to the global level.

Based on the timeline that was proposed by the World Health Organization [57], we set the *alert phase* from 31 December 2019 (when the WHO posed a media statement made by the Wuhan Municipal Health Commission on their website about the cases of ‘viral pneumonia’ in Wuhan, China PRC) to 11 March 2020 (when the WHO assessed that COVID-19 could be characterized as a pandemic).

Given that public health emergencies have a close relationship with human activities, the risk at the national level may differ from the one at the global level in particular ways. It was noticed that on 30 January 2020, the WHO declared the COVID-19 outbreak as a public health emergency of international concern (PHEIC), the WHO’s highest level of alarm [57]. Prior to this day, China accounted for 99% of the confirmed cases worldwide, while there were only 68 confirmed cases distributed across 15 countries other than China [58]. Therefore, we further divided the alert phase into two stages. *Stage 1* refers to the national level alert phase (from 31 December 2019 to 29 January 2020), during which the outbreaks primarily had a public health impact within the affected countries. *Stage 2* refers to the global level alert phase (from 30 January 2020 to 10 March 2020), during which the outbreaks became a public health risk to other countries and required a coordinated international response [11].

This study examined the global media citation network of COVID-19-related news at the two stages by analyzing the network structure characteristics, the composition of the core and periphery of the network, and the performance of network nodes. The research questions of the study are:How did the overall characteristics of the news media citation networks change from Stage 1 to Stage 2 of the COVID-19 alert phase?How did the core and periphery of the news media citation networks change from Stage 1 to Stage 2 of the COVID-19 alert phase?How did the key nodes of the news media citation networks change from Stage 1 to Stage 2 of the COVID-19 alert phase?

## 2. Method

### 2.1. Data Collection

This study collected data from the LexisNexis news database. The news report search formulas and selection process are presented in Figure 1. The LexisNexis news database collects reports from various data sources, including but not limited to newspapers, newswires, press releases, web news, news transcripts, and video news. We started with the denominations of the disease and virus to retrieve news reports about COVID from newspaper and newswire. There were many different denominations of COVID-19 and its causative virus by the health authorities, the news media, and the politicians in news reports, especially before the World Health Organization and the International Committee on Taxonomy of Viruses officially named them on 12 February 2020. The incorrect denominations created severe obstacles to data collection. We pre-reviewed 500 news reports and concluded that these denominations fall into two categories. One is a description of the disease, that is, the word “pneumonia” plus modifiers (e.g., atypical, mysterious, unexplained). The other is a description of the causative virus, that is, the word “coronavirus” or “virus” plus modifiers (e.g., deadly, lethal, pneumonia-like). In addition, many incorrect denominations and full texts of news reports include geographic locations that are associated with the origin of the disease (e.g., China, Wuhan, Chinese). Similar conclusions were reached by Prieto-Ramos et al. [59] and Dong et al. [60] on institutional and news media denominations of COVID-19 and its causative virus. Therefore, in this study, three Boolean retrievals were designed according to the denominations and geographical locations to avoid missing data. By randomly choosing and manually reviewing a sample of 500 cleaned reports, we detected two invalid data records and thus the error rate was 0.4%. Eventually, we constructed a corpus that contained 246,759 reports, of which 23,972 reports and 807 media outlets that covered news at Stage 1, while 222,787 reports and 1776 media outlets presented the news at Stage 2.

### 2.2. Constructing the Media Citation Network

This study used social network analysis to explore the relationships among members in the media citation network. A comprehensive analysis was conducted at three levels, namely the overall network, community, and node. The network was constructed based on the extracted media-to-media citation relationships. *First*, we created a thesaurus for retrieving the names of media outlets, including their official names that were presented in the LexisNexis news database and their abbreviations and aliases in their homepages and Wikipedia. *Second*, we extracted nodes from all the news reports and verified the textual context using both machine-based and manual methods to uncover relationships among media citation. *Third*, the obtained data were used to construct a media citation network. The nodes in the network represent media outlets; the direction of edges corresponds to the direction of information delivery; and the weights of edges represent the number of citations.

A total of 32,556 citation connections were obtained from 966 media outlets in this study. Compared with Stage 1, Stage 2 showed a significant increase in the corpus size. However, the proportion of reports that contain citations decreased from 16.41% to 12.84%. In addition, at Stage 2, media outlets were more likely to publish stories without citations. We constructed two media citation networks, Network-S1 and Network-S2, representing the network for Stage 1 and Stage 2, respectively. Network-S1 contained 414 nodes and 1211 edges, while network-S2 had 946 nodes and 4772 edges.

### 2.3. Measuring the Network Structure

In this study, we used the properties of overall metrics and typology to measure the network structure. We started by measuring the overall metrics (e.g., size, weight, density) of the two networks to understand the network structure. Then, we compared the topological properties of the two networks, ignoring the direction and weights of the edges. We examined whether the two networks were scale-free [61] by conducting a regression analysis of the node complementary cumulative degree distribution. The equation is as follows:


(1)
$$F\left(n\right)=a\cdot n^{-b}$$


In this equation, *n* denotes the complementary cumulative degree; *F*(*n*) is the probability density function of the variable *n*; *a* is the normalization constant; and *b* is the power exponent.

We examined whether the two networks had small-world characteristics [62] by calculating the average shortest path length and the local clustering coefficient. The average shortest path length is defined as the average number of edges in the shortest path between any two nodes in the network. The local clustering coefficient measures how close the neighbors are of a specific node are. The average local clustering coefficient refers to the average of the local clustering coefficients of all the nodes in a network.

### 2.4. Identifying and Analysing the Communities

In many cases, highly connected nodes, or nodes with the highest betweenness have little effect on a given propagation process [63]. In contrast, the most effective propagators are often those in the core of the network. Further, while those major components that are located on the network periphery lack influence on the rest of the network, they significantly impact local regions.

This study used the k-shell algorithm [63] to identify the core and periphery of the network. First, all the nodes with only one edge were removed together with their connected edges and then were placed in the 1-shell. The above process repeatedly went on until there were no such nodes in the network. Then, all the nodes with edge weights of 2 or less were removed before being placed in the 2-shell. This process was repeated until all nodes were assigned to the corresponding shell [64]. Overall, Network-S1 was partitioned into 15 shell layers, while Network-S2 was divided into 31 shell layers.

After completing the k-shell decomposition, the k-core (a graph consisting of all the nodes with a shell number that was greater than or equal to k) and the k-crust (a graph consisting of all the nodes with a shell number that was less than or equal to k) were generated [64]. Figure 2 shows the changes in size from the largest component to the second-largest component of different k-curst for both networks.

We found that at the points of $k_{1}=10$ (in Network-S1) and $k_{2}=11$ (in Network-S2), the k-crust’s largest component’s number of nodes increased significantly, while the second-largest component’s node number started to decrease, indicating that the k-shell of layers with corresponding or higher number of cores played the most important role in network connectivity. Therefore, we set the 10-core (43 nodes) and 11-core (192 nodes) as the cores of Network-S1 and Network-S2, respectively. The cores of networks play critical roles in information transmission in the network. The 9-crust (371 nodes) and 10-crust (754 nodes) were set as the peripheries of Network-S1 and Network-S2, respectively.

This study further analyzed the structures of the core and periphery. First, we identified the composition of the core, analyzed the core’s propagation coverage, and identified information flow within the core at the national level. Second, we identified the main components of the periphery and analyzed how they affected information dissemination among media outlets that were far from the core. In the previous step, two relatively large components with 67 and 32 nodes were identified in Network-S1 (the third largest component contained 12 nodes); the largest component of Network-S2 contained 82 nodes (the second-largest component contained 20 nodes). This study used the GN algorithm [65] components of the peripheral network to discover the most strongly connected media outlets. The GN algorithm seeks to identify cohesive subgroups by iteratively calculating the betweenness centrality of all the edges and removing the edge with the largest value. This study used the modularity Q to evaluate the community detection results as follows:(2)$$Q=\sum_{i}e_{ij}-a_{i}^{2}$$

In this formula, $e_{ij}$ denotes the proportion of edges whose two linking nodes are in the same community i, and $a_{i}$ denotes the proportion of edges in which at least one node is in community i to all edges in the network. Q is a value between 0 and 1, which converges 1 when a perfect community structure is detected.

### 2.5. Measuring Node Performance

There were two metrics, namely the level of activity and the level of influence, that were used to measure how powerful the media outlets are in terms of their influence on the spread of risk information in global media systems. This study defined the activity level as the total number of outgoing and incoming citations to and from a specific outlet in a citation network. We adopted the total degree (i.e., the sum of indegrees and outdegrees) to measure the activity. To ensure that the node degrees were comparable and to resist the influence of changes in network size, we normalized the two networks in this study. In the following formula, the *ndegree* denotes the *degree* of nodes after network normalization:(3)$$ndegree=\frac{degree}{number\;of\;nodes-1}$$

The influence of a given network node is measured by the change in other nodes that are influenced by that specific node [66,67]. This study used the PageRank algorithm [68] to measure the global influence of a media outlet in the citation network. The PageRank algorithm can be considered as an integration of the degree centrality method that calculates the weighted sum of edges with all the neighboring nodes and the closeness centrality method that calculates the shortest distance to reach other nodes. Compared to the above two methods, the PageRank algorithm has a clear advantage: when evaluating the overall influence of a specific outlet, it considers not only the number of citations but also the influence of the media outlets that are being cited. At the beginning, each node is given the same PR value (PR). After that, the current PR of each node is divided equally among all the nodes it points to. The above process is iterated until the PR with each node stabilizes.

Due to rank leak and rank sink, the PRs of some nodes eventually converge to 0. Therefore, a damping factor d is required to be introduced to deal with this problem. In the ranking algorithm for web pages, the parameter d usually takes a value of 0.85 [66], 1−d represents the probability that a surfer starts a new search without hopping to a neighboring node through a link. For example, supposing an editor or journalist who decides to use an external message, they may either read a report containing a citation and then find additional information sources by following the citation chain, or directly access the original report and cite it (either because of a collaborative agreement between media or by visiting a news site). Therefore, we defined the damping factor d as the probability of the former situation, that is, the proportion of randomly obtained reports that are not original. In global media systems, the value of the parameter d can be considered as the proportion of all reports that cite external messages. Based on data analysis from the entire observation period, the value of d that was derived in this study was 0.13.

Based on the introduction of parameter d, the PR of each node is equally distributed to all the nodes in the network with a probability of 1−d, before being distributed to the nodes it points to according to the edge weight with the probability of d. In each iteration step, the $PR\left(A\right)\;$ of node *A* was calculated as follows:(4)$$PR\left(A\right)=\frac{1-d}{N}+d\cdot \sum_{i}\frac{n_{T_{i}cA}\cdot PR\left(T_{i}\right)}{C\left(T_{i}\right)}$$

In this formula, $PR\left(T_{i}\right)$ denotes the PR of the previous iteration of node $T_{i}$ that is connected to node *A*; $C\left(T_{i}\right)$ is the times that node $T_{i}$ cites all nodes; $n_{T_{i}cA}$ is the times that node $T_{i}$ cites node *A*; and *N* is the total number of nodes in the citation network.

## 3. Result

### 3.1. Network Structure

**Overall characteristics.** The overall characteristics of the two networks are outlined in Table 1. Compared with Network-S1, Network-S2 has larger numbers of nodes, edges, and total weights but lower network density. The presumed reason is that more less-connected media outlets join Network-S2 compared with Network-S1, making the relationships in the Network-S2 relatively sparse. Both networks contained a maximally connected subgraph that contained the vast majority of each network’s nodes, and these subgraphs help COVID-19 risk information spread rapidly throughout global media systems.

Through aggregating the relationships among the media to the national/regional level (see Figure 3), it is apparent that from Stage 1 to Stage 2, the number of countries or regions to which the outlets belong increased, and the geographic scope of the networks expanded. Moreover, the countries/regions became more connected to each other.

**Topological characteristics of the network**. The regression analysis results of the node complementary cumulative degree distributions display the scale-free characteristics of both networks (see Figure 4). The complementary cumulative degree distributions followed a power-law distribution, suggesting that the networks of Stage 1 (Network-S1) and Stage 2 (Network-S2) are approximately scale-free networks as shown in Figure 4.

Figure 5 shows the distribution of shortcut distance between each network’s nodes. Small-world networks usually have smaller characteristic path lengths and higher clustering coefficients. Both Network-S1 and Network-S2 showed small-world characteristics. In both networks, most shortcut distances did not exceed 3. Additionally, the characteristic path lengths were 3.004 and 2.799, and the average local clustering coefficients were 0.434 and 0.523, respectively.

The media citation network also showed characteristics as follows: (1) Most media outlets had a small number of connections to each other, while a few outlets were more connected to other media, and they act as critical nodes of the media citation network. (2) 97% of the media could interact with other media through no more than four intermediaries, and thus the cost of information dissemination of the whole media system was low. (3) Some media formed collaborative cliques whose members interacted relatively closely and there were many shortcuts between these cliques. (4) Network-S2 showed a typical small-world nature with more shortcuts and cliques compared with Network-S1.

### 3.2. Network Core and Periphery

**Network core.** The core of the media citation network at Stage 2 was dramatically more enormous in scale than that of Stage 1. As shown in Table 2, the core of Network-S2 had 4.47 times more nodes and 8.88 times more connected edges than Network-S1. The core compositions of the two networks are presented in Figure 6 and Figure 7, respectively.

Figure 8 shows the composition and dissemination range of the cores. In terms of the geographic distribution of media outlets in the core, the share of outlets from the Americas in Network-S2 was roughly the same compared to Network-S1. The share of Europe and Oceania outlets decreased in Network-S2, while the share of Asian outlets increased in Network-S2. The African media have not joined the core until Stage 2 (the first case in Africa occurs at Stage 2). Among Asian media, Mainland China’s outlets and outlets of Hong Kong, Macao, and Taiwan (noting that these regions have the closest economic, cultural, and geographical proximity to Mainland China) occupied a significantly lower share of the core in Network-S2 than that in Network-S1. 

In terms of the core’s distribution coverage, most of the nodes were not in the core but were directly connected to the core, and the core played a crucial role in the spread of risk information. Media outlets within and connected to the core of Network-S1 accounted for 87% of the total media system, while those in the same network positions in Network-S2 accounted for 97%. At Stage 2, the network’s cores were relatively sparse but covered a much larger area. We also analyzed the data by continent and found that the geographical distribution of the media outlets in the core was uneven, especially in Network-S1, where Oceania outlets were more connected to the core, while the opposite was true for African media outlets.

Figure 9 presents the information flow between countries that are in the cores of the two networks. The rows in this figure refer to the countries/regions that cited foreign media, while the columns represent the countries/regions whose media were cited by foreign media. The countries/regions were sorted in rows and columns by the total number of citations, so that the hot spots were concentrated in the upper left corner of the charts. Darker colors in the graph represent more frequent citations among the outlets of two countries. In this figure, each country or region is denoted by two-letter country codes (see the Appendix A) when presenting the data that are related to specific countries or regions.

At Stage 1, media in the US and Canada cited foreign media significantly more often, while media reports from the US, Mainland China, France, Hong Kong, and Spain were cited more often by foreign media. At Stage 2, Canada cited foreign news significantly more often, followed by the US and the UK. Messages from the US media were cited significantly more often by foreign media, followed by France, the UK, and Mainland China. The media in some countries had a particular influence on that of another country. Typical examples included Kuwait on the UK, Spain on the US, and South Korea on Singapore. The largest number of citations between countries tended to occur between the US and Canada. Most of the information that was exchanged in the core was concentrated in a few countries.

**Network periphery.** The periphery of Network-S1 had two important components that contain six meaningful communities (Q = 0.765) as shown in Figure 10. Different colors represent different collaborative communities. The larger component on the right side of the figure can be divided into four subnets. The blue subnet contains all Oceania media in the figure, and the green subnet comprises mostly American media. These two components approximate a star structure, and their centers are *The Daily Telegraph* (Australia) and *Reuters* (UK). The gray subnet contains mostly Arab media. The orange subnet is suspended above the entire component by the edge “*Jordan News Agency* (Jordan)-*Middle East News Agency* (Egypt)” and includes media from several continents. The smaller component on the left side of the graph contains two subnets that are shown in red and yellow. The yellow subnet consists of media from the US and Commonwealth of Nations (e.g., the UK, Canada, India, and South Africa). The red subnet connects media from countries surrounding Mainland China (e.g., India, Pakistan, Japan, South Korea, Vietnam, Sri Lanka, Thailand, and Singapore). Its most important node is *Asia News Network* (Singapore). Some Arab media (i.e., *Jordan News Agency* (Jordan), *Middle East News Agency* (Egypt), *Arab News* (Saudi Arabia), *Khaleej Times* (United Arab Emirates)) played bridging roles between these subnets.

The most significant component in Network-S2′s periphery contains four significant subnets (Q = 0.584), as shown in Figure 11. The European media mainly make up the yellow subnet with the Gazette (UK) as the center and the orange subnet with the most internal connections. The red subnet contains all African media in this component and is connected to the other subnets through *The Bulletin* (US) and *The Advertiser* (Australia). All African media in this component are in the red subnet, which only receives information from *The Bulletin* (US) in the green subnet. The composition of the green subnet is diverse and sparse. The media community from Arab countries and the media community from countries neighboring Mainland China that exists in Network-S1 are not present in Network-S2. An inspection of the data reveals that most of these media outlets are in the core of Network-S2, which explains the greater proportion of Asian media participation in the core of Network-S2 to some extent.

### 3.3. Node Performance

**Level of activity.** Table 3 shows the level of activity (measured by the total ndegree) of the top 10 media outlets including US, Canada, France, the UK, Mainland China, and Hong Kong. *Associated Press, The Canadian Press, Agence France-Presse, ContentEngine, Xinhua News Agency*, *National Post,* and *MailOnline* presented a high level of activity in both stages of the pandemic alert phase. As outlets from the first outbreak country, *South China Morning Post* and *China Daily* had high activity only in Network-S1, reflecting the geographical correlation between media activity and the public health emergency.

For those media that have already presented in Network-S1, this study compared two networks to examine changes in their levels of activity (i.e., node activity/network-wide activity percentage). The results showed that from Stage 1 to Stage 2, *Kuwait News Agency* (Kuwait) (from 0.72% to 1.65%), *Associated Press* (US) (from 8.85% to 9.40%), and *The New York Times* (US) (from 1.79% (from 1.79% to 2.40%) saw the most significant increases in activity. In comparison, *Xinhua News Agency* (Mainland China) (from 4.18% to 2.31%), *South China Morning Post* (HK) (from 2.67% to 0.83%), and *China Daily* (Mainland China) (from 2.11% to 0.93%) witnessed the sharpest declines in activity. By aggregating media outlets to the country level, we found that UK media (from 12.87% to 15.82%), US media (from 26.45% to 27.69%), and Kuwaiti media (from 0.74% to 1.67%) showed the largest increases in activity across the network. Mainland China media (from 6.52% to 3.42%), Canadian media (from 21.14% to 18.18%), and Hong Kong media (from 2.77% to 0.86%) showed the largest decreases in their activity. 

**Level of influence.** Table 4 shows the top 10 nodes in terms of influence in both networks. Media outlets from the US moved up their ranking at Stage 2 and have been the most influence throughout the alert phase, followed by media outlets from Mainland China and the UK. *Associated Press*, *Xinhua News Agency*, *Press Association*, *Agence France-Presse*, *Cable News Network*, *South China Morning Post*, *The New York Times*, and *The Washington Post* had the most decisive influence in both networks. *Australia Broadcasting Corporation* (Australia) and *Financial Times* (UK) replaced *China Daily* (Mainland China) and *The Canadian Press* (Canadian) to be the most influential press at Stage 2.

For those media that were already present in Network-S1, this study compared their levels of influence (i.e., node influence/network-wide influence percentage) in both networks. The results showed that from Stage 1 to Stage 2, only *The New York Times* (US) (from 0.668% to 1.143%), *Cable News Network* (US) (from 0.789% to 0.944%), and *Australian Associated Press* (Australia) (from 0.292% to 0.295%) increased their influence across the networks, while all other media witnessed a decrease in influence. The above results indicate that a large number of media joining the network at Stage 2 diluted the influence of most media at Stage 1. The greatest decreases in influence were seen in *Xinhua News Agency* (Mainland China) (from 1.324% to 0.481%) and *China Daily* (Mainland China) (from 0.658% to 0.254%), indicating that as the pandemic spread from the outbreak countries to the globe, the influence of the media are closely related to the level of risk of the public health emergency. 

## 4. Conclusions

This study investigated the global media citation networks of COVID-19-related news in the *alert phase* of the pandemic and examined how the news flow changed when the alert phase escalated from the national level (Stage 1) to the global level (Stage 2). In particular, we analyzed the characteristics of the network structure, the composition of the network core and periphery, and the performance of network nodes. 

The findings reveal that the small-world pattern and scale-free property of media citation networks contributed to the rapid spread of COVID-19-related news around the world. Within the networks, a small number of media outlets from a few countries formed the backbone of the network to control the risk communication; meanwhile, many media of geographical and cultural similarities formed cross-border collaborative cliques in the periphery of the network. When the alert phase escalated from Stage 1 to Stage 2, the network demonstrated the following changes. 

*First*, more media outlets connected to the key nodes in shorter distance. The overall network density decreased and the connections between the nodes became relatively sparse, but the shortest distance between the nodes became smaller and more accessible. At the same time, the network’s core became more extensive, and the media that were directly connected to the core occupied a larger proportion of the entire media system. *Second*, the composition of the core changed along with the geographical position of the risk with the core getting larger when the pandemic spread to more countries in Stage 2. While the proportion of the core that was occupied by Asian media increased, the proportion of media from Mainland Chinese media, Hong Kong, Macao, and Taiwan media decreased significantly. The share of Europe and Oceania outlets decreased, and African media did not join the core until Stage 2. *Third*, the collaborative cliques moved from the periphery to the core of the networks, showing the globalization of risk communication to some extent. The cliques in Stage 1 (e.g., the Arab country’s media communities and the media community from countries neighboring Mainland China) and some key nodes of them (e.g., Reuters, The Daily Telegraph) enter the core in Stage 2. *Fourth*, while some key media outlets maintained their influence or importance during the alert phase, their influence was diluted when the alert phase escalated to the global level. In particular, the influence level and activity level of the media from Mainland China and Hong Kong declined significantly, indicating the association between the key nodes and the geographical location of health emergencies.

The findings of the study contribute to the understanding of news flow for risk communication in public health emergencies. Although there is a large body of research on risk communication of public health emergencies [19,29], few studies have investigated the news flow for risk communication from a network perspective. This study contributed to the literature by exploring the global media citation network of COVID-19-related news and revealing how global media systems (the core and periphery of the network) cooperated in risk communication. Second, this study contributed to the literature by considering the evolutionary nature of public health emergencies, i.e., revealing how the news flow changed when the alert phase escalated from the national level to the global level. This study also contributed to the understanding of international news flow by measuring the level of activity and level of influence of media outlets that may influence risk communication of global public health emergencies.

This study has some limitations. *First*, the spread of a pandemic disease often includes several phases. This study has focused on the alert phase and was not able to analyze the data for the interpandemic, pandemic, and transition phases of the COVID-19 pandemic. Future work should observe the complete cycle of the pandemic to explain the patterns of risk information transmission in the COVID-19-related public health emergencies. *Second*, when citing external sources, media journalists can be influenced by multiple factors (e.g., relationships between media outlets); therefore, the role of the damping factor d in media citation network analysis and its impact on the performance of the PageRank algorithm needs further investigation.

In short, this study attempted to unveil the law of information dissemination for risk communication in the alert phase of the COVID-19 public health emergency. We investigated the global media citation network of COVID-19-related news at the national level alert stage and the global level alert stage. The findings reveal that the small-world pattern and scale-free property of media citation networks contributed to the rapid spread of COVID-19-related news around the world. At the national level alert stage, a small number of media outlets from a few countries formed the backbone of the network to control the risk communication, while many media of geographical and cultural similarities formed cross-border collaborative cliques in the periphery of the network. When the alert phase escalated from the national level to the global level, the core became larger with collaborative cliques moving from the periphery to the core, but the influence of some key media outlets was diluted. The findings have important implications for global health inequities. At the onset of the pandemic, a relative lack of access to critical and timely public health information could contribute to a significant delay in the prevention and protection from the pandemic.

## Figures and Tables

**Figure 1 ijerph-19-09588-f001:**
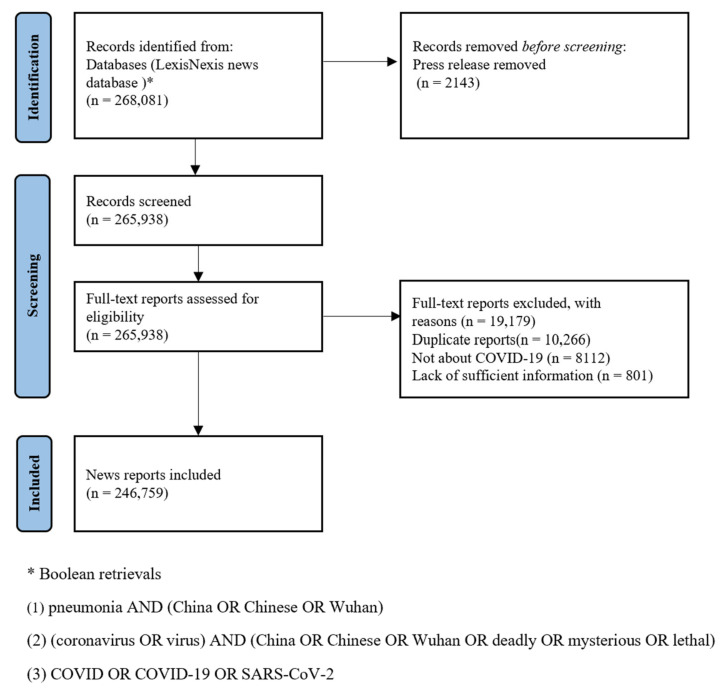
The news report search formulas and selection process.

**Figure 2 ijerph-19-09588-f002:**
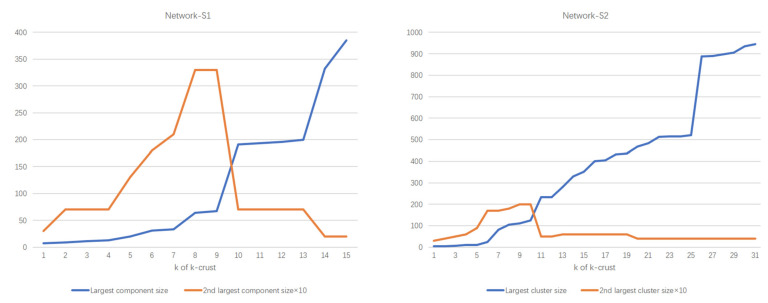
Size of the largest vs. second-largest component in k-crust.

**Figure 3 ijerph-19-09588-f003:**
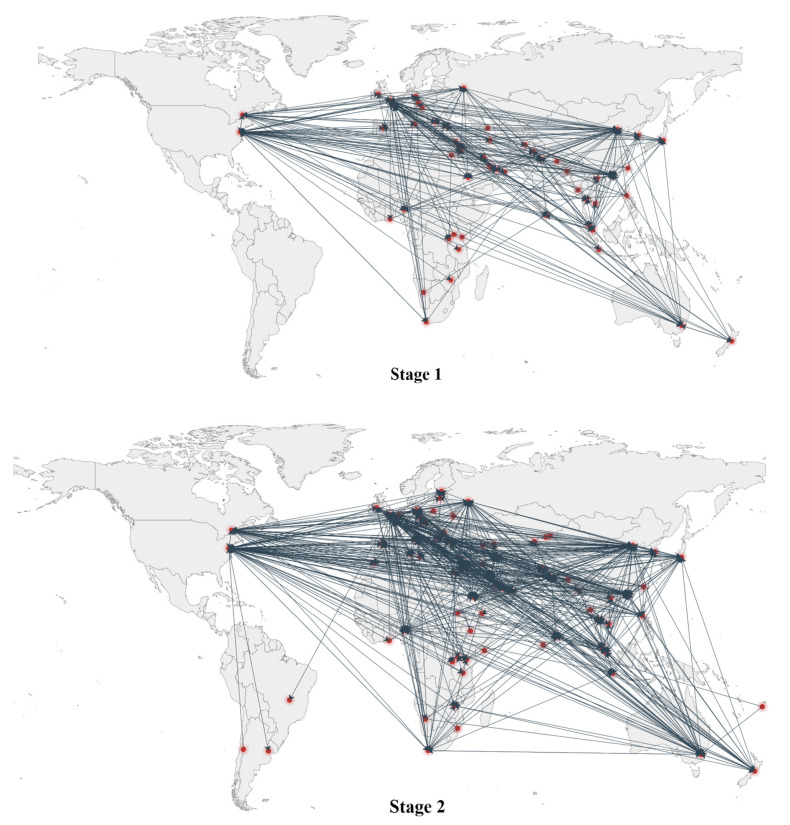
Media connections at the national/regional level.

**Figure 4 ijerph-19-09588-f004:**
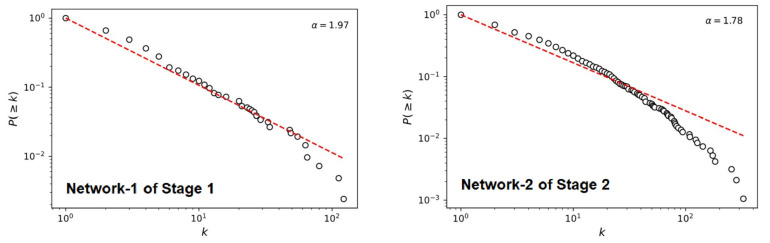
Complementary cumulative degree distribution of the two networks.

**Figure 5 ijerph-19-09588-f005:**
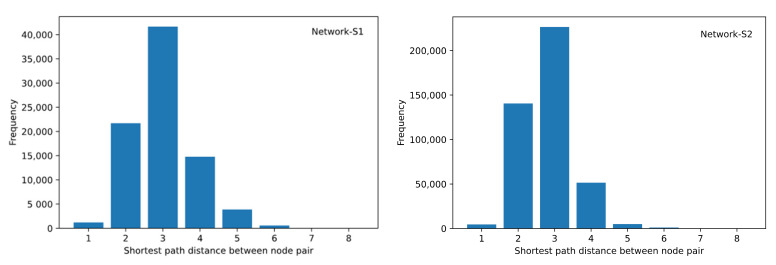
Shortcut distance distributions of the two networks.

**Figure 6 ijerph-19-09588-f006:**
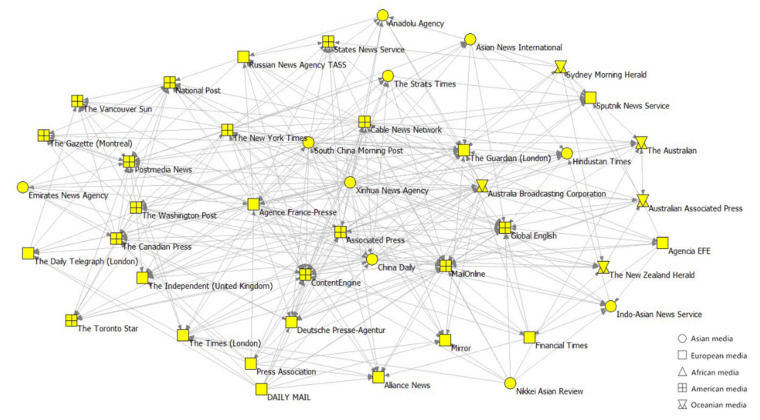
The core of Network-S1.

**Figure 7 ijerph-19-09588-f007:**
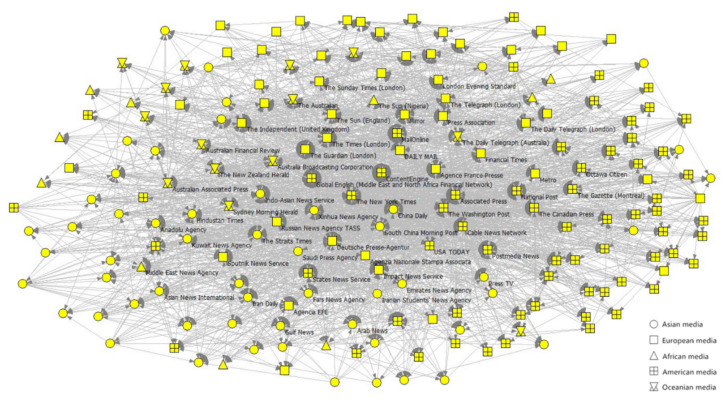
The core of Network-S2.

**Figure 8 ijerph-19-09588-f008:**
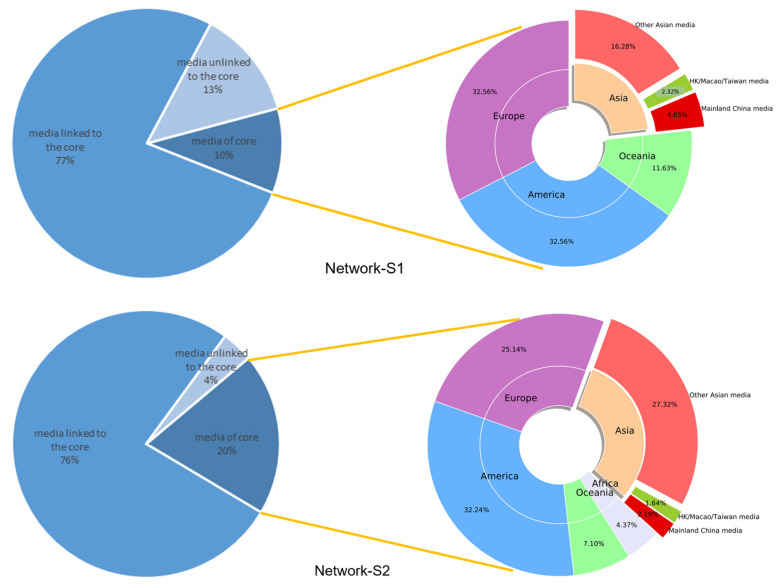
The composition and diffusion range of the cores.

**Figure 9 ijerph-19-09588-f009:**
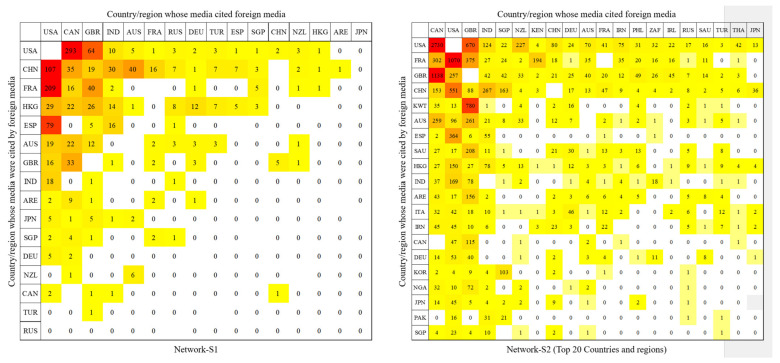
Media citations at the national level in the core.

**Figure 10 ijerph-19-09588-f010:**
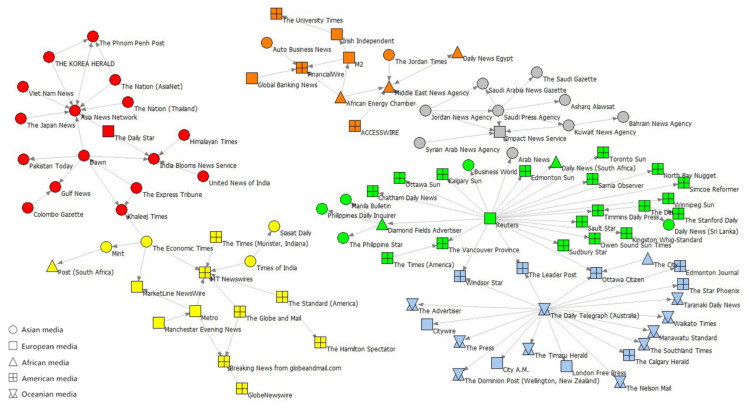
The main components of the edge of Network-S1.

**Figure 11 ijerph-19-09588-f011:**
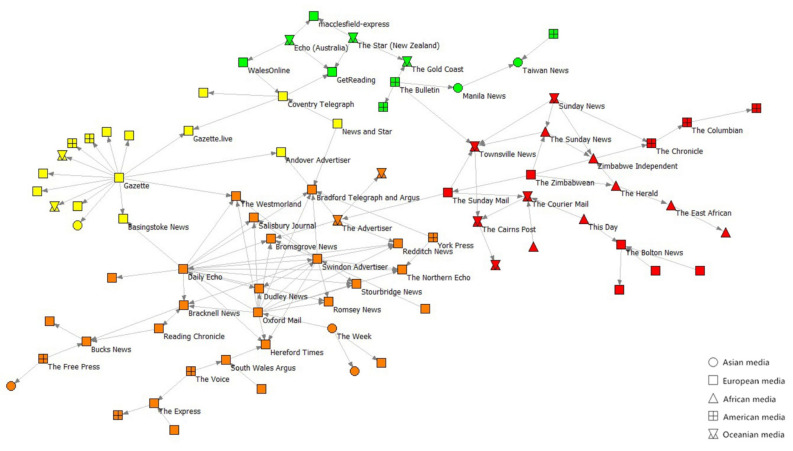
The main component of the edge of Network-S2.

**Table 1 ijerph-19-09588-t001:** Overall characteristics of the media citation networks.

Stage	Number of Nodes	Number of Edges	Total Edge Weights	Density	Number of Components	Largest Component Size
Stage 1	414	1211	3936	0.007	3	410
Stage 2	946	4772	28,620	0.005	9	927

**Table 2 ijerph-19-09588-t002:** Metrics for the Cores of the Two Networks.

Network	Number of the Core’s Nodes	Number of the Core’s Edges	The Average Local Clustering Coefficient of the Core
Network-S1	43	304	0.501
Network-S2	192	2699	0.494

**Table 3 ijerph-19-09588-t003:** Top 10 Media Outlets in Terms of the Level of Activity.

Network	Rank	Media	Country/Region	Total Ndegree
Network-S1	1	*Associated Press*	USA	1.627119
	2	*The Canadian Press*	Canada	1.271186
	3	*Agence France-Presse*	France	1.000000
	4	*ContentEngine*	USA	0.978208
	5	*Xinhua News Agency*	Mainland China	0.796610
	6	*South China Morning Post*	Hong Kong	0.508475
	7	*National Post*	Canada	0.469734
	8	*China Daily*	Mainland China	0.421308
	9	*Postmedia News*	Canada	0.418886
	10	*MailOnline*	Canada	0.411622
Network-S2	1	*Associated Press*	USA	5.698413
	2	*The Canadian Press*	Canada	3.680423
	3	*Agence France-Presse*	France	2.797884
	4	*ContentEngine*	USA	2.156614
	5	*The New York Times*	USA	1.452910
	6	*Xinhua News Agency*	Mainland China	1.400000
	7	*Impact News Service*	United Kingdom	1.355556
	8	*National Post*	Canada	1.299471
	9	*MailOnline*	Canada	1.131217
	10	*Cable News Network*	USA	1.087831

**Table 4 ijerph-19-09588-t004:** Top 10 Media Outlets In Terms of Influence.

Network	Rank	Media	Country/Region	PageRank
Network-S1	1	*Associated Press*	USA	0.016915
	2	*Xinhua News Agency*	Mainland China	0.013245
	3	*Press Association*	United Kingdom	0.011630
	4	*Agence France-Presse*	France	0.007972
	5	*Cable News Network*	USA	0.007893
	6	*South China Morning Post*	Hong Kong	0.006989
	7	*The New York Times*	USA	0.006682
	8	*China Daily*	Mainland China	0.006580
	9	*The Canadian Press*	Canada	0.006110
	10	*The Washington Post*	USA	0.005043
Network-S2	1	*Associated Press*	USA	0.013947
	2	*The New York Times*	USA	0.011430
	3	*Cable News Network*	USA	0.009437
	4	*Press Association*	United Kingdom	0.008192
	5	*Xinhua News Agency*	Mainland China	0.004814
	6	*Agence France-Presse*	France	0.004381
	7	*The Washington Post*	USA	0.004038
	8	*South China Morning Post*	Hong Kong	0.003378
	9	*Australia Broadcasting Corporation*	Australia	0.003355
	10	*Financial Times*	United Kingdom	0.003268

## Data Availability

The authors declare that all data supporting the findings of this study are available within the article.

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
