# Peer review of "Risk Communication in the Alert Phase of the COVID-19 Pandemic: Analysis of News Flow at National and Global Levels"

_ijerph, 2022, doi:10.3390/ijerph19159588_

Round 1
Reviewer 1 Report
Overall comments. This study offers a novel approach to quantitatively examine how risk information flow occurs at the national and global levels during the early stages of global health emergencies, from a network perspective. The authors used social network analysis to examine how international news media coverage of foreign countries may facilitate information exchange process about the SARS-COV-2 virus. Importantly, results on the study provide insights into how changes in the global health news exchange aligned with changes in public health emergencies. While results provided in this paper is substantial, discussion of the implications of these findings is relatively lacking. I recommend accepting this manuscript for publication with revisions.
Introduction
Line 69: What do you mean by “regular” risk coverage? Do you mean diseases that is prevalent, like flu, which has been around for a long time? Or do you mean diseases with low severity, like cold, which is common?
Line 74: I am confused about the description of “large” media. Do you mean countries tend not to report public health issues in foreign countries? Wording of
Line 95-96. For those unfamiliar with the theory, please provide a brief description of core, semi-periphery, and periphery zones to help understand this paragraph.
Line 111: Translational news isn't a commonly used terminology in communication. Could authors define translational news and provide an example of transnational coverage from the cited study?
Line 121: unequal and equal in what? Do you mean Western countries received more news coverage in foreign news media than Asian news countries did in foreign news channels?
Line 128: I think there is a typo in new production, this needs to be corrected to “news”.
Methods
Information is needed to justify the choices of search terms for all three formulas. Typically, such search is based on the literature and terminology/thesaurus from the library. Also, the paper could benefit from including a visual diagram to describe the systematic process of screening/selecting the papers included in the study, such as the PRISMA flow diagram used in the systematic reviews paper.
Line 196: Why was pneumonia included for Formula A?
Lines 202-204 on the LexisNexis should be moved to line 194-195 right after introducing the dataset, at the beginning of Data Collection.
Lines 204-205: This sentence: “We chose newspapers and newswires for our study, as these media usu-204 ally contain more credible news sources.” I think this justification is weak, as "usually more credible news source" is open to interpretation. e.g., how do you conceptualize credible, and what is the frequency of "usually"? I recommend removing this sentence. However, I'd be interested in seeing justification over why the authors included press releases, since they are not typically from the neutral third-party media.
Results
Line 364: Figure 3 is missing labels for each graph (which is Network 1 and Network 2?)
Line 396: What do you mean by “basically unchanged?” Do you mean it had similar patterns?
Line 398: I think “what’s more” should be removed.
Line 421: The Figure 8 provides a strong visual representation of the density of the core nations’ media citations. I’m curious why the size of two charts are different? Should they be re-scaled to match the size? Also, the order of the countries in Network S1 and S2 are inconsistent (Why is CAN second in S1 but first in S2? I think there needs justification for this or change in the order for consistency to help the readers.
Also, this table needs additional row on the top to label the row (country citing the media) and column (country receiving citation)
Line 429: what do you mean by “always”? Is it possible that the occurrence was 100%? If not this wording should be removed.
Figure 9: What do each of these colors represent? Do they represent different continents or do they serve specific roles pertaining to the media distribution?
Line 462: Fig 10 is missing.
Line 508: Table 5 is missing.
Discussions and Conclusions.
Compared to the results, authors provide relatively little discussions of these findings. I recommend the authors to expand upon each of their key findings described in the results section. In lines 521-527, the authors provide four key highlights that are easy to follow. Further interpretations are needed to justify each of the four points made in this paragraph.
Line 527: This sentence: “In particular, the importance of the media from Mainland China and Hong Kong declined significantly, indicating the association between the key nodes and the geographical location of health emergencies.” What do you mean by “importance”? This study measured distribution and scope of the media coverage, not individual perceptions of the importance of media. I don’t think such conclusion is valid.
Over the past years, we have seen how COVID-19 disparately and systematically affected developing nations, further widening global health disparities. The results show that the Western news media had a domineering presence over information flow compared to developing nations. I believe that these results have important implications for global health inequities, such that relatively lack of access to such critical and timely public health information at the onset of the pandemic could have contributed to a significant delay in prevention and protection from COVID-19 in these nations, compared to the Western countries, such as the US and UK. I believe that this implication is relevant and timely to help improve the value of this study findings.
Line 551: After the limitations, authors need to provide A summary paragraph to recap the study goal, results, and implications for conclusion.
Author Response
Dear Editors and Reviewers,
We appreciate your time reviewing our paper and giving very helpful comments and suggestions, which have guided us to improve the quality of this paper. We have carefully read all the comments and addressed them in the revised manuscript. A point-by-point revision report is attached for easy reference.

Reviewer 2 Report
I appreciate your intellectual contribution to the topic of COVID-19 pandemic, which shows us the characteristics of the news flow of COVID-19 pandemic at different phases, though there are a few errors:
In Line 29, is that "information dissertation" or information "dissemination"?
In Line 129, it should be foci (or alternative words with similar meanings).
Table 5 is missing on page 508.
Finally, it seems that the WHO declared the COVID-19 outbreak as a public health emergency of international concern (PHEIC) on Jan. 30, 2020. However, you wrote "Jan. 30, 2019" in Line 171. You may double-check the information.
Overall, thank you for expanding my understanding of this topic from the perspective of network analysis of news flow across the world.
Author Response

(The authors gave the same response as above.)
